# Ultrastaging of the Parametrium in Cervical Cancer: A Clinicopathological Study

**DOI:** 10.3390/cancers15041099

**Published:** 2023-02-09

**Authors:** Nicolò Bizzarri, Damiano Arciuolo, Camilla Certelli, Luigi Pedone Anchora, Valerio Gallotta, Elena Teodorico, Maria Vittoria Carbone, Alessia Piermattei, Francesco Fanfani, Anna Fagotti, Gabriella Ferrandina, Gian Franco Zannoni, Giovanni Scambia, Denis Querleu

**Affiliations:** 1UOC Ginecologia Oncologica, Dipartimento per la Salute della Donna e del Bambino e della Salute Pubblica, Fondazione Policlinico Universitario A. Gemelli, IRCCS, 00168 Rome, Italy; 2Dipartimento per la Salute della Donna e del Bambino e della Salute Pubblica, Gynecopathology and Breast Pathology Unit, Fondazione Policlinico Universitario A. Gemelli, IRCCS, 00168 Rome, Italy; 3Istituto di Clinica Ostetrica e Ginecologica, Università Cattolica del Sacro Cuore, 00168 Rome, Italy

**Keywords:** cervical cancer, parametrium, ultrastaging, radical hysterectomy, micrometastasis, lymph node

## Abstract

**Simple Summary:**

The rate of positive parametrial lymph nodes in a series of apparent early-stage cervical cancer patients undergoing bilateral sentinel lymph node mapping was 3.1%. After ultrastaging of parametrial tissue, no occult micrometastasis was detected.

**Abstract:**

Occult parametrial involvement in apparent early-stage cervical cancer might be overlooked with standard pathologic assessment. The primary endpoint of the present study was to assess the rate of positive parametrial lymph nodes and of microscopic continuous or discontinuous parametrial involvement. This is a retrospective, single-center, observational study including patients with FIGO 2018 stage IA1–IIA1 and IIIC1p in whom bilateral sentinel lymph node (SLN) detection and ultrastaging of SLN were performed according to institutional protocol, with surgery as primary treatment performed between May 2017 and February 2021, as well as type B2/C1/C2 (Querleu–Morrow) radical hysterectomy and usual histology (squamous cell, adenocarcinoma and adenosquamous carcinoma). Thirty-one patients were included in the study period. Six (18.7%) patients had metastatic lymph nodes, of whom four had only SLN metastasis (two cases of ITC, one case of micrometastasis and one case of macrometastasis). We found a macroscopic deposit of cancer cells in the parametrial lymph node of one patient (3.1%). There was a positive statistical correlation between the incidence of parametrial lymph node involvement and the metastatic pelvic lymph nodes (*p* = 0.038). When performed per patient, the sensitivity, negative predictive value and accuracy of parametrial lymph node involvement in predicting pelvic lymph node metastasis were 16.7%, 83.3% and 83.9%, respectively. Ultrastaging of parametrial tissue did not identify any occult continuous or discontinuous parametrial metastasis. In conclusion, the incidence of lymph node parametrial involvement in a retrospective series of early-stage cervical cancer was 3.1% of all included patients. Lymph node involvement of the parametrium was associated with lymph node metastasis. The sensitivity of parametrial lymph node involvement to predict pelvic lymph node metastasis was low. The lack of parametrial involvement revealed by parametrial ultrastaging could be related to the number of patients with tumors with a pathologic diameter < 2 cm (54.8%). Further prospective studies are needed to analyze the role of parametrial ultrastaging in early-stage cervical cancer and to assess whether it can be considered the “sentinel” of the sentinel lymph node.

## 1. Introduction

Lymph node metastasis is one of the major prognostic factors in early-stage cervical cancer [1]. Multiple studies have described a correlation between tumor characteristics and parametrial involvement [2,3], as well as between parametrial and lymph node involvement [4]. Previous pathology studies described the characteristics of parametrial involvement in cervical cancer [5,6]. In the sentinel lymph node (SLN) era, one recent study demonstrated a strong relation between parametrial involvement and SLN metastasis after SLN ultrastaging [7]. More recently, one study focused on the analysis of upper paracervical lymphovascular tissue, which was subjected to ultrastaging and immunohistochemistry (IHC) [8]. Interestingly, the authors showed that metastatic paracervical lymph nodes were found in 2.4% of patients without other pelvic lymph node metastasis. Nevertheless, to the best of our knowledge, no study in the literature performed ultrastaging of the entire parametrial tissue in patients who underwent SLN ultrastaging.

The primary endpoint of the present study was to assess the rate of metastatic parametrial involvement (lymph nodes or microscopic continuous/discontinuous). Secondary endpoints were to assess the correlation between pelvic lymph node metastasis and microscopic parametrial involvement, as well as the recurrence rate and the pattern of recurrence.

## 2. Materials and Methods

### 2.1. Patients Characteristics

This is a retrospective, single-center, observational study. The study was approved by the Policlinico Agostino Gemelli IRCCS Ethical Committee on 21 April 2022 (ID 4851, protocol number 0014772/22). We included consecutive patients with histological diagnosis of cervical cancer treated at Policlinico Universitario Agostino Gemelli IRCCS meeting the following criteria: FIGO 2018 stage IA1–IIA1 and IIIC1p, bilateral SLN detection, ultrastaging of SLN performed according to institutional protocol [9], surgery as primary treatment, type B2/C1/C2 (Querleu–Morrow) radical hysterectomy [10], usual histology (squamous cell, adenocarcinoma or adenosquamous carcinoma) and surgery performed between May 2017 and February 2021. All patients underwent a preoperative staging MRI scan and a pelvic US scan. Minimally invasive surgery and laparotomy were performed until the publication of the LACC trial [11], followed by laparotomy thereafter. Patients were submitted to SLN biopsy with or without pelvic lymphadenectomy according to international guidelines [12]. SLN mapping was performed with indocyanine green in all cases (both in the case of minimally invasive and open approaches [13]). Patients assessed for SLN with one-step nucleic acid amplification or with hematoxylin and eosin (H&E) only, with evidence of gross continuous/discontinuous parametrial involvement upon H&E staining [5] or undergoing fertility-sparing procedures were excluded. Non-SLNs were examined with routine H&E staining. A modification of the American Joint Committee on Cancer staging definitions for axillary lymph nodes in breast cancer was used to classify metastatic disease (macrometastases defined as tumors greater than 2.0 mm in diameter; micrometastases defines as tumor cell aggregates between 0.2 and 2.0 mm in diameter; isolated tumor cells (ITCs) defined as individual tumor cells or aggregates less than 0.2 mm in diameter with fewer than 200 cells) [14].

### 2.2. Ultrastaging of Parametrial Tissue

Paraffin-embedded specimens (sampled at time of surgery and named “parametrium” or “paracervix”) from included patients were retrieved from a pathology archive, and 3-dimensional lateral parametrial tissue ultrastaging was performed with 150-micron slices until exhaustion of the tissue (Figure 1). The nomenclature of lateral parauterine tissue, usually named “lateral parametrium” in surgical language, has been the subject of discussion in previous publications [15,16]; although it can be defined as paracervical/parauterine tissue, in the present study, we refer to it as “parametrium” for simplicity. Immunohistochemistry analysis of ultrastaged parametrial tissue was conducted with antibodies anti-cytokeratin AE1:AE3. A dedicated gynecologic oncology pathologist reviewed all obtained slides.

### 2.3. Statistical Analysis

Standard descriptive statistics were used to assess variables. The group with metastatic lymph nodes was compared with the group with negative lymph nodes; continuous variables were compared with Student’s *t*-test, and categorical variables were compared with the chi-square test. Sensitivity, negative predictive value and accuracy of parametrial lymph node involvement were calculated taking the pelvic lymph node metastasis as the referral standard (isolated tumor cells were considered metastatic pelvic lymph nodes, although they have been shown to have no prognostic significance [17]). All *p* values reported herein are two-sided, and a *p* value < 0.05 was considered statistically significant. 

Analysis was computed using SPSS version 27.0 (IBM Corporation 2018, Armonk, NY, USA).

## 3. Results

A total of 222 patients underwent tracer injection for SLN mapping in the study period. The exclusion process is demonstrated in Figure 2. Thirty-one (14.0%) patients were included. Table 1 shows patients characteristics. Six (18.7%) patients had metastatic lymph nodes, of whom four had only SLN metastasis (two cases of ITC, one case of micrometastasis and one case of macrometastasis), one had SLN and non-SLN metastasis and one had non-SLN metastasis only (false-negative SLN). 

We found one metastatic parametrial lymph node in one patient (3.1%). In the analysis per hemipelvis, metastatic parametrial lymph nodes were detected in one of seven (14.3%) sidewall with any type of pelvic lymph node involvement (also including low-volume metastases, i.e., ITCs and micrometastases) and in one of five (20.0%) cases with macroscopic and microscopic metastatic lymph nodes only. There was a positive statistical correlation between the presence of parametrial lymph node involvement and metastatic pelvic lymph nodes (*p* = 0.038) (characteristics of patients with and without lymph node metastasis are reported in Table 2). The patient with positive parametrial lymph nodes was diagnosed with a grade 3 squamous cell carcinoma with a pathologic diameter of 21 mm and lymph vascular space involvement. She had one macrometastasis in an obturator SLN and one macrometastasis in a right-side pelvic non-SLN. She received adjuvant chemoradiotherapy and was alive and free of disease at 42-month follow-up. 

Ultrastaging of the parametrium of all patients generated 660 slides, which were analyzed with IHC; no occult continuous or discontinuous parametrial metastasis was identified. Table 3 shows the characteristics of patients with metastatic lymph nodes and the outcome of parametrial involvement. 

When performed per patient, the sensitivity, negative predictive value and accuracy of lymph node parametrial involvement in predicting pelvic lymph node metastasis were 16.7%, 83.3% and 83.9%, respectively (Appendix A).

With a median follow-up time of 24 months (95%CI: 18.5–29.4), four (12.9%) patients experienced recurrence, and none of them died. Two (50.0%) recurrences were located in the vaginal vault, and two (50.0%) were identified as distant metastases. One (25.0%) of the recurrent patients had metastatic lymph nodes at the time of radical hysterectomy.

## 4. Discussion

In the present study, we aimed to assess the rate of lymph node parametrial involvement and the role of parametrial ultrastaging in patients treated with radical surgery for apparent early-stage cervical cancer. We showed that lymph node parametrial metastasis was detected in 3.1% of patients. The incidence of lymph node parametrial involvement was evident in 14.3% of hemipelvis samples with all positive lymph nodes and in 20.0% of hemipelvis samples with macro- and micro-metastases in pelvic lymph nodes. The sensitivity of parametrial lymph node involvement in predicting pelvic lymph node metastasis was found to be 16.7%. Ultrastaging of parametrial tissue with IHC did not detect any occult continuous or discontinuous parametrial metastasis.

The present work was designed to investigate the relation between parametrial and pelvic lymph nodes (including SLN) and whether ultrastaging of parametrial tissue could detect in-transit cancer cells toward the pelvic sidewall, even in cases with negative pelvic lymph nodes. Previous studies have analyzed this issue with different methodologies; pathology analyses performed with giant sections of radical hysterectomy specimens revealed potential metastasis of the parametrial lymph nodes, described as “continuous, vessels, discontinuous or lymph node parametrial involvement” [5,18]. The same pathology technique was subsequently adopted by Benedetti-Panici et al., who found positive parametrial nodes in 12% of stage IB–IIA cases [19], and later by Winter et al., who found parametrial involvement in 8.9% of patients with clinical stage IB and in 10% of patients with stage IIA (they included only patients with negative pelvic lymph nodes) [20]. Recently, the SENTICOL study group analyzed the incidence of parametrial involvement in the era of SLN and found that in patients with <20 mm tumors on preoperative MRI scan and negative SLN after ultrastaging, the incidence of parametrial involvement was only 0.9%, suggesting a two-step procedure to triage patients to decide whether to perform radical versus non-radical hysterectomy [7]. This is in line with our results, as the only patient with positive parametrial tissue (lymph node) was a patient with a tumor >20 mm and with metastatic SLN. Moreover, it is also known that in patients with very low-risk tumors (squamous histology, <20 mm, preoperative negative pelvic node status upon MRI scan), the risk of lymph node involvement is extremely low [21]. 

Recently, Lührs and colleagues showed that 52.4% of early-stage cervical cancer patients had at least one “parauterine” lymph node identified when subjected to ultrastaging and IHC [8]. Moreover, they identified metastatic “paracervical” lymph nodes in 2.1% of all women and 15.8% of node positive women without lateral pelvic lymph node metastases. Metastatic “parauterine” lymph nodes were found in 2.4% patients without other pelvic lymph node metastases. We have to acknowledge a few differences between the abovementioned study compared to our study. First, Lührs’ study was prospective, while ours is retrospective; this might represent a major bias, as sampling of specimens in the retrospective cases was performed years before, and the entire parametria may not have been embedded in paraffin blocks. Secondly, the “paracervical lymphovascular tissue” in their study was isolated at the time of surgery by gynecologic oncologists and sent as separate tissue for pathological investigation. Third, we performed ultrastaging of the entire tissue, which was archived as “parametrium”/“paracervix”, while Lührs intraoperatively identified exclusively the tissue lateral to the broad ligament, medial to the umbilical artery, proximal/caudal to the supravesical artery and ventral to the ureter with the aid of ICG. Despite these differences, the overall incidence of parametrial/paracervical lymph node involvement in the two series was similar (2.1% versus 3.1% in all patients). Nevertheless, they found a positive “parauterine” node in 2.4% patients without other pelvic lymph node metastases, which was not the case in our series. This is a relevant finding, as it represents an indication for adjuvant chemoradiotherapy in patients who otherwise would have been only followed-up. Interestingly, the presence of metastatic parametrial/paracervical nodes as unique lymph node involvement poses the classification dilemma of staging as FIGO 2018 IIB versus IIIC1p, highlighting the need for a consensus on the most recent FIGO staging [22,23].

The ultrastaging of parametrial tissue did not identify any microscopic occult continuous or discontinuous parametrial metastasis upon IHC, leading to the rejection of our initial hypothesis and highlighting the importance of thorough analysis of parametrial lymph nodes rather than parametrial connective/vascular tissues.

The present study is subject to some limitations. First, as mentioned, the retrospective nature of its design and potential bias in selection of parametrial tissue sampled and archived at the time of surgery represent limitations. Secondly, the fact that the majority of patients had tumor diameters <20 mm (54.8%) and were LVSI-negative (58.1%), representing a group at low risk of parametrial involvement according to previous literature [2,3]. On the other hand, we are not aware of previous studies submitting entire sampled parametrial specimens for ultrastaging.

## 5. Conclusions

The incidence of parametrial lymph node involvement in a retrospective series of apparent early-stage cervical cancer was 3.1% of all included patients and 16.7% of patients with metastatic pelvic nodes. Parametrial ultrastaging was not found to be a tool suitable for upstaging the disease, as no occult microscopic metastasis was found. Lymph node involvement of the parametrium was associated with lymph node metastasis. The sensitivity of parametrial node involvement to predict pelvic lymph node metastasis was low. Further prospective studies are needed to analyze the role of parametrial ultrastaging in early-stage cervical cancer and to assess whether it can be considered the “sentinel” of the sentinel lymph node.

## Figures and Tables

**Figure 1 cancers-15-01099-f001:**
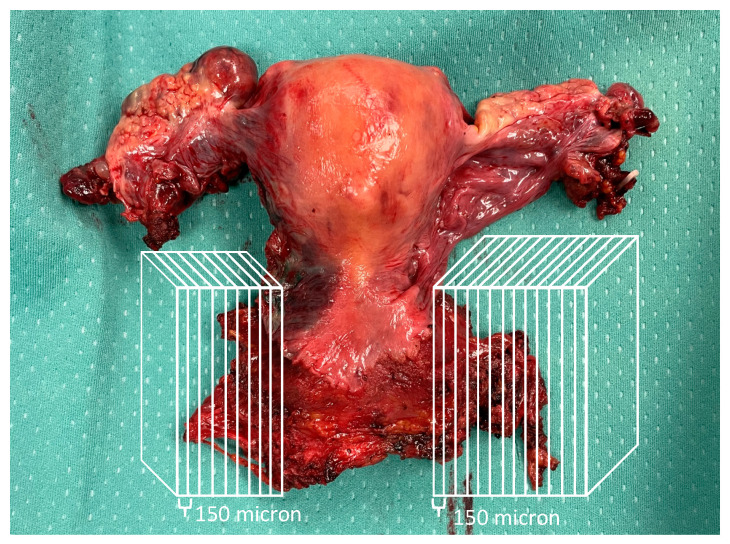
Example of how parametrial ultrastaging was performed (three-dimensional paraffin-embedded blocks were sliced at a distance of 150 microns).

**Figure 2 cancers-15-01099-f002:**
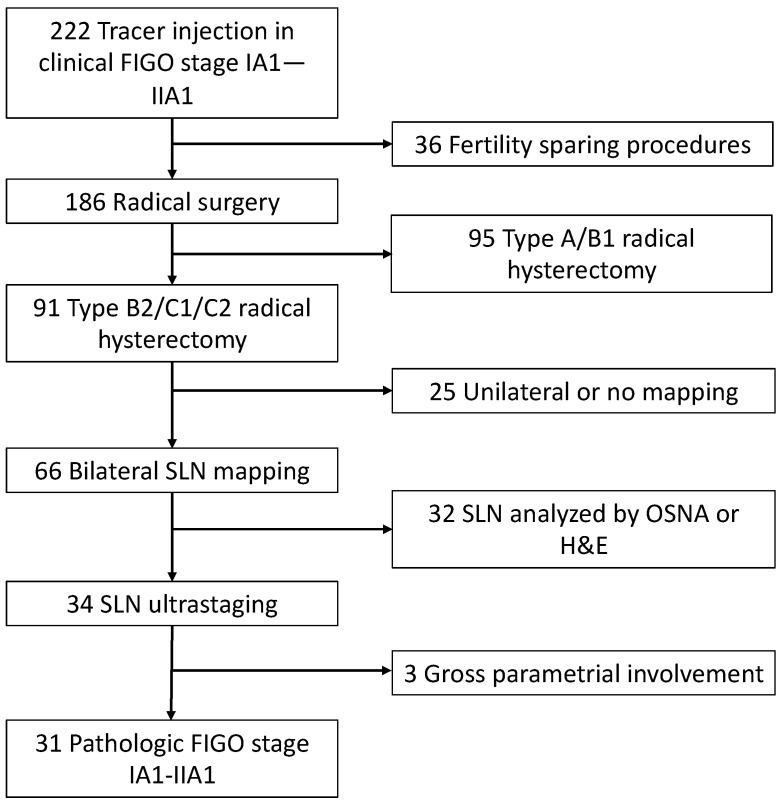
Flow chart showing inclusion and exclusion process.

**Table 1 cancers-15-01099-t001:** Patient characteristics.

Characteristic	N = 31 N (%), Median (Range)
Age, years	44 (27–68)
BMI, kg/m^2^	24 (17–35)
FIGO 2018 stage	
IA1	6 (19.4)
IA2	3 (9.7)
IB1	8 (25.8)
IB2	7 (22.6)
IB3	2 (6.5)
IIA1	1 (3.2)
IIIC1p	4 (12.9)
Histology	
SCC	20 (64.5)
Adenocarcinoma	11 (35.5)
Surgical approach	
Laparotomy	14 (45.2)
Laparoscopy	12 (38.7)
Robot	5 (16.1)
Radicality of hysterectomy (Querleu–Morrow)	
B2	11 (35.5)
C1	20 (64.5)
Lymph node assessment	
SLN only	5 (16.1)
SLN + pelvic lymphadenectomy	26 (83.9)
Histological tumor diameter (mm)	17 (1–60)
LVSI	
Negative	18 (58.1)
Positive	11 (35.5)
Unknown	2 (6.5)
Grading	
Well-differentiated	1 (3.2)
Moderately differentiated	17 (54.8)
Poorly differentiated	7 (22.6)
Unknown	6 (19.4)
Depth of stromal infiltration	
<50%	19 (61.3)
≥50%	12 (38.7)
Pathologic tumor diameter	
<2 cm	17 (54.8)
≥2 cm	14 (45.2)
Metastatic pelvic lymph nodes	
SLN only	4 (12.9)
SLN and non-SLN	1 (3.2)
Non-SLN only	1 (3.2)
Volume of lymph node metastasis *	
ITC	2 (6.4)
Micrometastasis	1 (3.2)
Macrometastasis	3 (9.7)
Adjuvant treatment	
No	18 (58.1)
Radiotherapy	8 (25.8)
Chemoradiotherapy	5 (16.1)

* In patients with more than one metastatic node only the largest volume metastasis has been reported.

**Table 2 cancers-15-01099-t002:** Comparison of characteristics of patients with and without metastatic pelvic lymph nodes (ITC and micro- and macrometastasis).

Characteristic	Negative Lymph Nodes (N = 25) N (%), Median (Range)	Metastatic Lymph Nodes (N = 6) N (%), Median (Range)	*p* Value
Age, years	44 (27–68)	47 (34–55)	0.822
BMI, kg/m^2^	24 (17–35)	25 (20.5–27)	0.781
Histology SCC Adenocarcinoma	15 (60.0) 10 (40.0)	5 (83.3) 1 (16.7)	0.383
Surgical approach Laparotomy Minimally invasive	12 (48.0) 13 (52.0)	2 (33.3) 4 (66.7)	0.664
LVSI Negative Positive Unknown	16 (64.0) 7 (28.0) 2 (8.0)	2 (33.3) 4 (66.7) 0	0.103
Grading Well-differentiated Moderately differentiated Poorly differentiated Unknown	1 (4.0) 16 (64.0) 5 (20.0) 3 (12.0)	0 1 (16.7) 2 (33.3) 3 (50.0)	0.278
Depth of stromal infiltration <50% ≥50%	16 (64.0) 9 (36.0)	3 (50.0) 3 (50.0)	0.653
Pathologic tumor diameter <2 cm ≥2 cm	14 (56.0) 11 (44.0)	3 (50.0) 3 (50.0)	0.791
Parametrial lymph nodes Negative Positive	25 (100) 0	5 (83.3) 1 (16.7)	0.038

**Table 3 cancers-15-01099-t003:** Characteristics of patients with metastatic lymph nodes.

Tumor Characteristics	SLN	Non-SLN	Parametrial Involvement
Stage: IB1, Diam:20, LVSI: pos, Hist: SCC, DOI: ≥50%, G3	R: macro L: macro	R: negative L: negative	Negative
Stage: IB1, Diam:21, LVSI: pos, Hist: SCC, DOI: ≥50%, G3	R: macro L: negative	R: macro L: negative	Positive (lymph node)
Stage: IA2, Diam:5, LVSI: neg, Hist: SCC, DOI: <50%, Gx	R: ITC L: negative	R: negative L: negative	Negative
Stage: IB1, Diam:7, LVSI: pos, Hist: SCC, DOI: <50%, G1	R: negative L: micro	R: negative L: negative	Negative
Stage: IA1, Diam:2, LVSI: neg, Hist: ADC, DOI: <50%, Gx	R: negative L: ITC	R: negative L: negative	Negative
Stage: IIA1, Diam:21, LVSI: pos, Hist: SCC, DOI: ≥50%, G2	R: negative L: negative	R: macro L: negative	Negative

R: right sidewall; L: left sidewall; LVSI: lymph vascular space involvement; SCC: squamous cell carcinoma; DOI: depth of stromal infiltration; ADC: adenocarcinoma.

## Data Availability

The data presented in this study are available upon request from the corresponding author.

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
