# Peer review of "Ultrastaging of the Parametrium in Cervical Cancer: A Clinicopathological Study"

_cancers, 2023, doi:10.3390/cancers15041099_

Round 1
Reviewer 1 Report
Congratulations to the authors! A very nice, clear and well written retrospective unicentric analysis looking at parametrial involvement and the use of ultrastaging in a small cohort of Cx patients with a predominantly low-risk population. I have nothing to object to the content, the data presented, the discussion or the conclusion. It's just a pity that the cohort is not larger and more rich in high-risk, larger tumors. It would have been a more interesting read. But even the authors themselves mention this as a limitation of their work.
Author Response
Thanks for this feedback. We are planning a further study involving a larger number of patients at higher risk.
Reviewer 2 Report
With pleasure, I read the paper titled: “Ultrastaging of parametrium in cervical cancer: a clinico-pathological study”. The article is clinically relevant and within the scope of the journal. Overall, the article reads well, the English language is proper, citations are adequate, and flow of ideas is smooth. The presented summaries in the form of tables are big bonus. The major strength is being the first-ever study to examine the impact of ultratrastaging of the entire parametrial tissue in patients who underwent SLN ultrastaging. The authors did a great work by comparing and contrasting their findings with the published literature. The limitations of the study are well-acknowledged. Overall, the authors revealed a low rate of parametrial lymph node involvement. The major weakens is the small number of sample size, which severely affects the power of the analysis and conclusions; nevertheless, this shortcoming was properly acknowledged.
Author Response
Thanks for this comment. We are planning a new prospective study involving a larger number of patients at higher risk.
Reviewer 3 Report
According to my opinion further, patients women should be included in multicentric studies with suitable inclusions criteria and evaluated the advantages and disavantages of parametrial ultrastaging. Further prospective studies long time are needed to analyze the role of parametrial ultrastaging in early-stage cervical cancer and to assess if it can be considered the “sentinel” of the sentinel lymph node.
Author Response
Thanks for your remark. We are planning a new prospective study involving a larger number of patients at higher risk in a multicentric setting.
Reviewer 4 Report
This is a retrospective study on the ultrastaging analysis of the lateral parameter in early-stage cervical cancers.
The argument is very interesting and a real topic of debate. Unfortunately this study cannot give us any conclusions, but it is surely the starting point for a dedicated prospective study.
The interest of ultrastaging on the parameter would be to predict the usefulness of hysterectomy enlargement, compared to simple hysterectomy. With only 1 positive lymph node in the parameter, this study cannot demonstrate a correlation between the parameter and the sentinel lymph node. Future studies should demonstrate the correlation between the positive parameter and the sentinel lymph node.
Author Response
Thank you for the comment. We are planning a new prospective study involving a larger number of patients at higher risk in a multicentric setting.